# Teaching methods for critical thinking in health education of children up to high school: A scoping review

Anna Prokop-Dorner[1]*, Aleksandra Piłat-Kobla[1], Magdalena Ślusarczyk[2], Maria Świątkiewicz-Mośny[2], Natalia Ożegalska-Łukasik[3], Aleksandra Potysz-Rzyman[4], Marianna Zarychta[4], Albert Juszczyk[5], Dominika Kondyjowska[5], Agnieszka Magiera[6], Małgorzata Maraj[7], Dawid Storman[7], Sylwia Warzecha[7], Paulina Węglarz[7], Magdalena Wojtaszek-Główka[5], Wioletta Żabicka[5], Małgorzata M. Bała[8]

1 Department of Medical Sociology, Chair of Epidemiology and Preventive Medicine, Jagiellonian University Medical College, Kraków, Poland, 2 Institute of Sociology, Jagiellonian University, Kraków, Poland, 3 Institute of Intercultural Studies, Jagiellonian University, Kraków, Poland, 4 LIGHT Project, Institute of Sociology, Jagiellonian University, Kraków, Poland, 5 Medical Faculty Student's Research Group for Systematic Reviews, Jagiellonian University Medical College, Kraków, Poland, 6 Department of Epidemiology, Chair of Epidemiology and Preventive Medicine, Jagiellonian University Medical College, Kraków, Poland, 7 Department of Hygiene and Dietetics, Chair of Epidemiology and Preventive Medicine, Jagiellonian University Medical College, Kraków, Poland, 8 Chair of Epidemiology and Preventive Medicine, Jagiellonian University Medical College, Kraków, Poland

☯ These authors contributed equally to this work.
* anna.prokop@uj.edu.pl

## Abstract

According to the World Health Organization, the improvement of people's health literacy is one of the fundamental public health challenges in the 21st century. The key issue in teaching health literacy is to develop critical thinking skills. As health literacy and critical thinking should be developed at school age, we reviewed teaching methods or educational interventions used in empirical studies focused on the development of critical thinking regarding health and implemented by teachers in preschools, primary schools, or secondary schools. We searched seven databases (Medline, Embase, Web of Science, ERIC, ProqQuest, PsycArticles, and CINAHL) from inception to 20 September 2023 for any type of empirical studies. Due to the heterogeneity in interventions and inadequate reporting of results, a descriptive synthesis of studies was performed in addition to quantitative analysis. Of the 15919 initial records, 115 studies were included in the review. Most of the educational interventions focused on lifestyle-related health issues such as substance use, sexual and reproductive health, and nutrition. The popularity of health issues changed over time and depended on the geographical context. Six dimensions that differentiated the teaching methods were identified: central teaching component, central educator, pupils' activity level, teaching context, educational materials, and significance of critical thinking. Many educational interventions did not address the development of critical thinking skills in a comprehensive manner, and the significance of critical thinking varied greatly. Interventions in which critical thinking had high and very high significance applied mainly problem-solving methods and involved pupils' activity. The evidence on the effectiveness of the teaching

**Data Availability Statement:** All relevant data are within the manuscript and its Supporting Information files.

**Funding:** This work is the result of research project Diagnosis and developing health capital - Health Literacy of primary school students (Project no. UMO-2020/39/B/HS6/00977) funded by the National Science Centre. The funders had no role in study design, data collection and analysis, decision to publish, or preparation of the manuscript. The protocol of the review was registered in the OSF Registries on 13 January 2022 (doi: https://doi.org/ 10.17605/OSF.IO/46TEZ).

**Competing interests:** The authors have declared that no competing interests exist.

methods that develop critical thinking is limited because most articles failed to provide detailed information on the teaching methods or did not examine their effects. We recommend that a checklist is developed to facilitate a detailed description of health educational interventions and thus promoting their replicability.

**Study registration:** The protocol of the review was registered in the OSF Registries on 13 January 2022 (doi: https://doi.org/10.17605/OSF.IO/46TEZ).

## Introduction

One of the major public health challenges in the 21st century is to improve people's health literacy [1]. Health literacy refers to an individual's ability to seek, understand, and use health information. Health literacy skills are essential for claim evaluation, data interpretation, and risk assessment. The key issue in learning health literacy is to develop knowledge, skills, motivation, and self-awareness that translate into individuals' autonomy, independence, and empowerment. These qualities enable individuals to deal with health and its determinants.

In its definition of health literacy, the World Health Organization stresses the importance of social competences, such as communication and critical thinking, which are necessary for making adequate health decisions both on daily basis [2] and in extraordinary circumstances, such as the pandemic [3]. The fundamental goal of acquiring health literacy is to develop critical thinking skills. Critical thinking means that people are able to analyze and evaluate their thought processes in order to improve them [4]. According to a widely used definition, critical thinking is "a reasoned, reflective thinking focused on deciding what to believe or do" [5]. Today, we live in a world of information, and critical thinking skills can help us think logically and clearly. The competence of critical thinking is essential because it allows people to think independently.

Considering the abundance of easily available, but not verified, information as well as global health threats such as the coronavirus disease 2019 (COVID-19) pandemic, critical thinking skills become especially important in such life domains as health [3]. People need these skills to critically assess and use information relevant to their health, and it is the key to make evidence-based health choices. For example, the COVID-19 pandemic can be viewed not only as a public health threat but also as an infodemic [6], because there was overabundance of fake news, misinformation, and conspiracy theories that have undermined the trust in health institutions and treatment procedures [7–32]. Machete et al [33] conducted a systematic review including 22 articles that were synthesized and used as evidence to determine the role of critical thinking in identifying fake news. The study confirmed that critical thinking skills are essential to recognize fake news.

In this context, it seems crucial to teach critical thinking to pupils (i.e., children up to high-school level). Fostering critical thinking is widely recognized as an integral part of developing health literacy. There are several strategies that are recommended for teaching critical thinking, including classroom discussions [34], problem-based learning [35], and questioning techniques [36, 37]. There is also evidence that peer-to-peer interaction is one of the teaching behaviors related to student gains in critical thinking [38]. However, most of these recommendations are based on theoretical works or do not relate to health-related topics. Moreover, these works refer to higher-education students, including students in a specific field (such as nursing or economics).

In this scoping review, we focused on the concept of health literacy and critical thinking as one of its main dimensions. We aimed to identify and review the teaching methods or pedagogical interventions used in empirical studies on the development of critical thinking regarding health and implemented by teachers in preschools as well as primary or secondary schools (level of education 0, 1, 2, and 3 according to the International Standard Classification of Education [ISCED]). The article presents the methods used in this process, quantitative and qualitative results, discussions of the findings, and conclusions.

## Materials and methods

We conducted the scoping review in accordance with the Joanna Briggs Institute [39] methodology for scoping reviews and in our reporting we adhered to the PRISMA (Preferred Reporting Items for Systematic Reviews and Meta-Analyses) reporting statement with extension for scoping reviews [40]. We provided the filled-out checklist in S1 Table. In the development of our review we followed the methods outlined in the protocol registered in the OSF Registries on 13 January 2022 [41].

### Criteria for study inclusion

For this scoping review, we considered any type of qualitative and quantitative empirical studies focusing on the development of critical thinking within the framework of health education at school by teaching subjects with content related to health (biology, chemistry, science, physical education, wellness, sexual education, health education, digital education, math, and critical thinking as a subject). Moreover, we included studies that provided information about teaching methods, training activities, or pedagogical interventions implemented by teachers or other school educators. Finally, we considered empirical studies referring to pupils in preschool, primary (elementary) or secondary (high) schools (ISCED 0, 1, 2, 3) and to teachers from those schools.

### Search strategy

We searched the following databases: Medline, Embase, Science Citation Index with Abstracts, ERIC, ProqQuest, PsycArticles, and CINAHL.

We employed the text words contained in the titles and abstracts of relevant articles, and the index terms used to describe the articles, to develop a full search strategy for each database (see S2 Table). We used the following terms in the key search strategy: "health knowledge", "health education", "health literacy", "critical thinking", "schools", "education", "informed choice", "choice behaviour", "decision making", "curriculum", and "teaching methods". We adapted the search strategy, including the relevant keywords and index terms, for each included database and/or information source. We screened the reference list of all included sources of evidence for additional studies. We searched databases from inception to 20 September 2023. Due to limited resources, we only included studies in English.

### Study selection and data collection

Following the search, we collated all identified citations, uploaded them into Endnote X8 (Clarivate Analytics, PA, USA), and screened using the Covidence online tool (covidence.org). We removed any duplicates using Covidence.

We performed the three rounds of calibration exercises, using 50 abstracts each downloaded into an MS Excel spreadsheet (which ensured a common understanding of the inclusion and exclusion criteria). Next, 14 authors (MMB, MŚM, MŚ, APK, APD, NO, DS, APR,

MZ, PW, WŻ, MM, SW, DK) working independently and in pairs screened the studies with respect to meeting eligibility criteria based the titles and abstracts. Thus, we obtained the full texts of potentially eligible articles. After four rounds of calibration exercises using five full texts each, 10 authors (MMB, DS, PW, SW, DK, MM, WŻ, APK, MWG, APD) working independently and in pairs screened the studies with respect to meeting eligibility criteria using their full texts. Third reviewer (MMB) resolved disagreements arising at any stage of the study selection. The core team developed and piloted the extraction form in Excel (MMB, MSM, MŚ, APD, APK, APR, MZ), and following four rounds of calibration exercises, eight reviewers (MM, SW, PW, DK, WŻ, AJ, MWG, AM) worked in pairs to extract data from the included studies into the prepiloted form. The pairs of reviewers independently extracted the data. Due to heterogeneity in interventions and inadequate reporting of results, we performed a descriptive synthesis of studies. The extracted data included specific details about the study methods, context (e.g., type of school, school location, study population), interventions, description of teaching methods focusing on critical thinking, and key findings relevant to the objectives of this review. Three authors (MMB, APD, APK) additionally checked all extractions.

## Qualitative data synthesis

To further analyze the teaching methods, we conducted a qualitative synthesis [42]. Based on the primary analysis of the extracted data, two authors (APD and APK) developed and tested a coding book in MAXQDA 2024 based on 5% of the included articles. We resolved any discrepancies in coding at this stage by discussion. We used the final coding book to code detailed information on the teaching methods and the practical strategies of their implementation provided in the articles and in external sources such as further publications or websites of the interventions. The process of summarizing and comparing the coded data as well as using graphical tools to identify patterns allowed us to precisely categorize the teaching methods into analytical themes (six dimensions of teaching methods). These themes were developed from free codes and descriptive themes.

## Results

A total of 15919 records of 15909 studies were initially identified. After removing duplicates, 15150 studies were screened on the basis of the title and abstract. This yielded 1056 potentially relevant studies, which were screened based on full texts. Of the 1056 studies, 243 (25.5%) were excluded because they did not concern the development of critical thinking. Other studies were excluded because they were only theoretical (n = 174), did not concern the population of interest (n = 171), did not address health literacy (n = 132), did not provide information about the teaching methods used (n = 99), or for other reasons (n = 116). The list of the excluded studies, along with reasons for exclusion, is available on the project website in the OSF Registries [17]. We identified 118 eligible studies, of which 3 were still ongoing [43–45]. Finally, we included 115 completed studies (Fig 1).

The included studies met the eligibility criteria and described the teaching methods used, but most of them (80%) did not examine the effectiveness of these teaching methods but interventions used in the study. Below we present the findings first referring to the quantitative and then to qualitative analysis.

## Description of the included studies

A total of 115 studies were included in this scoping review, including 65 studies reporting quantitative methods [46–113], 25 studies reporting mixed methods [114–140], and 25 studies reporting qualitative methods [7–32] (See S3 Table). Some educational interventions were

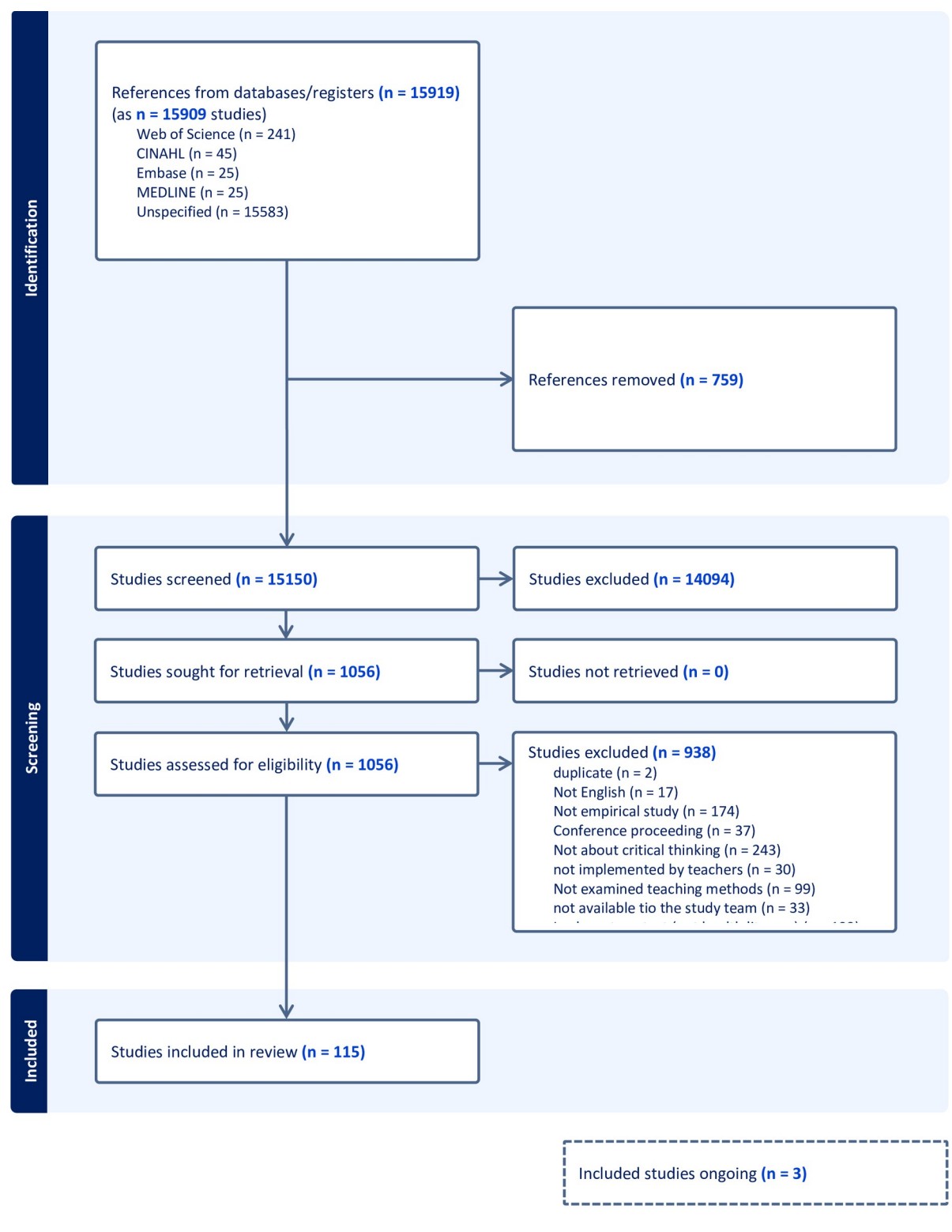

**Fig 1. Flow diagram on the selection of studies.**

described in more than one article. In such cases, the records were merged and assessed as one study [16, 17, 56, 70–72, 119]. The most common study design was cluster randomized (25 articles, 22%) and quasi-experimental (20 articles, 17%). The dates of article publication covered nearly 40 years. More than a half of the eligible articles were published after 2010 (74 articles, 64%) and only 12 studies were published before 2000 (10%). The included studies were conducted in various cultural contexts, but mostly in the Western societies of North America (52 articles, 45%) and Europe (34 articles, 30%). Only 14 studies were conducted in Asia (12%); 8, in Africa (7%); 5, in Australia (4%); and 2, in South America (2%). In one article, there was no information on the country [137].

Educational interventions conducted in North America covered a broad range of topics and addressed psychoactive substance use [21, 26, 50, 52, 53, 58, 61, 65, 67, 75, 80, 83, 85, 92, 95, 117, 140], lifestyle (including nutrition, physical activity) [57, 60, 63, 77, 87, 89, 96, 100, 135], sexual and reproductive health (SRH) [19, 49, 82, 94, 98, 108, 120, 127, 128] (including AIDS and HIV prevention [21, 59, 73, 86, 93]), public health [18, 31, 66, 69, 78, 79, 87, 90, 111], and somatic health [25, 87, 123, 131, 140]. The topic of mental health has only emerged in publications from the last three years [100, 104, 138].

Most studies conducted in Europe concerned lifestyle, including both nutrition and/or physical activity interventions [7, 9, 11, 22, 24, 46, 91, 97, 103, 106, 109, 126, 134], public health [8, 12, 13, 29, 47, 88, 101, 105, 139], and psychoactive substance use [7, 15, 48, 51, 84, 114, 122]. Four papers concerned somatic health [22, 30, 97, 125] and five–mental health [68, 97, 99, 109, 113]. Only two educational intervention addressed sexual health [28, 115].

Most studies conducted in Asia addressed sexual health [14, 56, 119, 132, 136, 141], including AIDS and HIV prevention [56, 116, 119, 133, 136]. Mental health was addressed by three studies [64, 112, 141], psychoactive substance use by two [74, 84]; and somatic health by one study [121]. In the last three years, studies have emerged whose educational interventions focused on lifestyle [27, 110]. Among African studies reporting on educational interventions, there were six articles that focused on SRH [10, 55, 62, 118, 124, 132], and one intervention that was dedicated to health claims [130].

Finally, research conducted in Australia concerned such health topics as psychoactive substance use [70–72, 81], lifestyle [16, 17], as well as public [23, 102] and mental health [23], while an educational intervention conducted in South America covered the topic of SRH [20].

## Health issues in education interventions

Interventions reported in the included articles addressed a broad range of health issues, and the thematic focus of the interventions had changed over time (Table 1). Until 2000, the prevailing topics in health education were substance use and SRH, in the following decades also

**Table 1. Health issues addressed in the tested interventions.**

| Decade of publication | Health issues | | | | | | |
|---|---|---|---|---|---|---|---|
| | Psychoactive substance use | SRH | Nutrition | Public health | Physical activity | Somatic health | Mental health |
| up to 1990 | 2 | 1 | 0 | 0 | 0 | 0 | 0 |
| 1991–2000 | 3 | 3 | 2 | 3 | 1 | 1 | 0 |
| 2001–2010 | 12 | 8 | 6 | 8 | 3 | 2 | 3 |
| 2011–2020 | 9 | 13 | 10 | 8 | 6 | 4 | 3 |
| from 2021 | 5 | 6 | 9 | 8 | 4 | 4 | 7 |

The number of publications calculated in rows. The colors indicate a relative number of publications calculated in the rows, with red indicating the highest and blue the lowest number. Only the most important health issues covered in the interventions were coded.

nutrition, issues connected with public health, physical activity, as well as somatic and mental health gained interested of teachers and stakeholders in the field.

Almost one in three studies published over the last 40 years tested substance use interventions (27%). Half of them discussed nicotine [50, 51, 53, 58, 61, 67, 70, 74, 75, 80, 83, 85, 95, 114] and drugs [21, 26, 52, 53, 65, 70–72, 74, 76, 81, 84, 85, 87, 140] and four in ten concerned alcohol [15, 48, 53, 71, 72, 83, 85, 92, 117, 122, 137].

The same number of interventions (31; 27,5%) covered SRH, and specifically sexual health [10, 28, 55, 78, 112, 115, 132], reproductive health [20, 49, 62, 82, 94, 115, 120, 127, 136], sexual abstinence [20, 49, 55, 73, 82, 124, 127, 128, 136], contraceptive methods [62, 94, 120, 124], menstruation [14], gender roles [32, 108, 119, 132], healthy relationships [32, 55, 119], sexually transmitted disease [82, 86, 118, 119], and AIDS and HIV prevention [21, 55, 56, 59, 73, 82, 86, 116, 118, 119, 124, 133, 136].

Slightly less studies (27; 23,5%) tested an intervention on nutrition (23,5%) [8, 11, 16, 17, 22, 23, 46, 57, 60, 77, 87, 89, 91, 96, 97, 103, 109, 110, 112, 123, 125, 126, 135, 140–142]. Public health problems, such as health care [21], violence [13, 18, 78], global health [8], organ donation [88], anti-microbial resistance [107], zoonosis [101], use of medicine [12], and bioethical dilemmas linked to health [47], social inequalities [31] were taught in 25% of reported interventions. Various forms of physical activity were promoted in every tenth intervention (11%) [7, 9, 16, 17, 63, 77, 87, 89, 97, 135, 142].

Specific somatic health issues such as cancer, cardiovascular system, diabetes, eye or oral health were discussed in 11% of the articles [22, 25, 30, 64, 87, 97, 112, 121, 125, 131]. Even fewer articles reported interventions on mental health issues, such as emotional regulation [64, 89, 97], resilience [23] and healthy relationships [23, 111, 119]. Nearly every third tested intervention covered more than one health issue [7, 16, 17, 21–23, 50, 55, 64, 77, 78, 82, 86, 87, 89, 97, 109, 111, 112, 119, 124–127, 135, 136, 140, 141]. Topics such as epidemic or pandemic were discussed only in a few articles, mainly with regards to HIV and AIDS [73, 116, 133] or social inequality during the COVID-19 pandemic [31]. Vaccinations were discussed in interventions generally linked to infectious disease [66] or aimed at increasing the uptake of specific vaccination, i.e. HPV [102].

Interventions reported in 94 articles (82%) were initiated by external bodies, such as universities, and were tested in several schools in a selected region (Table 2). Nearly half (51) of the studies tested regionally based interventions. In 31 studies, the interventions were tested locally, typically in one or in several schools. The remaining interventions were evaluated in bigger samples, either on a national (16 articles) or international level (5 articles). Nine of the interventions were pilot interventions. Moreover, the studied interventions varied in terms of the level of education. Most of them were tested in high schools/secondary schools (60, 52%); 30, in primary/elementary schools (26%); 24, in middle schools (21%); and only 1 intervention was tested in preschools. Interventions were conducted by schoolteachers, peer educators, or both. Half of the studied interventions were preceded by teachers' training (57 articles, 50%) and/or peer leaders training (13 articles, 11%). Only every third intervention provided pupils with additional materials, such as booklets [22, 32, 74, 77, 102, 124], handouts [49, 78, 117], audiovisual materials [20, 74, 90, 99, 107, 115], textbooks [84, 85, 130], recipes [57] newsletters [28, 46], exercise book [129], and student guide [111].

Interventions tested in the included articles were typically taught in class (50%), most often in an interdisciplinary form as part of multiple school subjects, such as health education or sexual health education, math, family life education, social sciences, media literacy, language, philosophy, home economy, science, and, less typically, during a single subject such as health education (23 articles), biology (3 articles), science (3 articles), sexual health education (3 articles), language (2 articles), critical thinking (1 article), social sciences (1 article), math (1

**Table 2. Characteristics of educational interventions in health.**

| Study ID | Reach of intervention | Type of school | Who initiated intervention | Subject | Teacher training | Peer training | Additional materials for pupils | Assessment of intervention effect | Details of teaching method |
|---|---|---|---|---|---|---|---|---|---|
| **Aghazadeh 2020** | Pilot | Elementary/ Primary school | External body | Math, Science, Language arts | Y | N | Y | Y | Y |
| **Anderson 2005** | Local | Elementary/ Primary school | External body | NR | Y | N | Y | Y | Y |
| **Alekseeva 2015** | National | Elementary/ Primary school and High school/ Secondary school | External body | NR | Y | Y | N | Y | Y |
| **Allsop 2022** | Regional | High school/ Secondary school | External body | Health education | Y | N | N | Y | N |
| **Arauz Ledezma 2021** | Local | High school/ Secondary school | External body | NR | Y | N | Y | Y | Y |
| **Araujo 2017** | National | High school/ Secondary school | External body | Biology, philosophy | Y | N | N | N | N |
| **Audrey 2006** | Regional | High school/ Secondary school | External body | Math, science, literacy, social studies | N | N | N | Y | Y |
| **Aventin 2020** | National | High school/ Secondary school | External body | Humanities/ social sciences, math | Y | N | Y | N | Y |
| **Banas 2021** | Local | High school/ Secondary school | Unclear | Unclear | N | N | N | N | N |
| **Basen-Engquis 1997, Coyle 1999** | Regional | High school/ Secondary school | External body | Unclear | Y | Y | Y | Y | Y |
| **Bell R 1993** | Regional | High school/ Secondary school | Unclear | Health education | N | N | N | Y | N |
| **Bell M 2005** | Regional | Elementary/ Primary school | External body | Unclear | Y | N | N | Y | N |
| **Begoray 2009** | Regional | High school/ Secondary school | External body | Health education | N | N | Y | Y | N |
| **Bond 2004** | Regional | High school/ Secondary school | External body | English, Health, Personal development | N | Y | Y | Y | N |
| **Bonnesen 2023** | National | High school/ Secondary school | External body | Danish, Social Studies, Physical Education and Sport, Introduction to Natural science | Y | N | N | Y | Y |
| **Borawski 2009** | Regional | High school/ Secondary school | External body | Health education, school nurses | Y | N | N | Y | Y |
| **Brinez 2019** | Local | Middle school | Internal body | Biology | N | N | N | Y | Y |
| **Brotman 2013** | Regional | High school/ Secondary school | External body | Health education, Science, English | Y | N | N | Y | N |
| **Bruselius-Jensen 2014, 2017** | Regional | Elementary/ Primary school | External body | Math | Y | N | Y | Y | Y |
| **Bruselius-Jensen 2017** | International | Elementary/ Primary school | Mixed | NR | N | N | Y | Y | Y |
| **Byers 2003** | Regional | Middle school | Already existing in the curriculum | Sexual health education | N | N | N | NR | N |
| **Caria 2011** | International | Middle school | External body | NR | Y | N | N | Y | Y |
| **Carlsson 2012** | International | High school/ Secondary school | External body | NR | N | N | N | Y | Y |
| **Carolan 2007** | Regional | Middle school | Unclear | Unclear | N | Y | N | Y | Y |

*(Continued)*

**Table 2.** (Continued)

| Study ID | Reach of intervention | Type of school | Who initiated intervention | Subject | Teacher training | Peer training | Additional materials for pupils | Assessment of intervention effect | Details of teaching method |
|---|---|---|---|---|---|---|---|---|---|
| Cheng 2008 | Regional | High school/ Secondary school | External body | NR | Y | NR | NR | Y | Y |
| Contento 2007 | Local | Middle school | Unclear | Science | Y | NR | N | Y | Y |
| Cooper 2022 | Regional | Elementary/ Primary school | External body | NR | Y | N | N | Y | N |
| Davis 2023 | Regional | High school/ Secondary school | External body | NR | Y | N | N | Y | Y |
| Dela Fuente-Anuncibay 2023 | Regional | Elementary/ Primary school | External body | NR | NR | N | N | Y | Y |
| Denny 2006 | Unclear | Upper elementary, middle school and High school/ Secondary school | External body | Health education | Y | NR | Y | Y | Y |
| DiCicco 1984 | National | High school/ Secondary school | External body | Health education, Science | Y | N | N | Y | Y |
| Dinaj-Koci 2015 | Local | High school/ Secondary school | External body | Health education | Y | N | N | Y | Y |
| Dunton 2012 | Regional | Elementary/ Primary school | External body | NR | N | N | N | Y | Y |
| Fage-Butler 2019 | Regional | Elementary/ Primary school | External body | Critical thinking | N | NR | NR | Y | Y |
| Flay 1985 | Regional | Middle school | External body | Health education | N | N | N | Y | Y |
| Ghimire 2020 | Local | Middle school | External body | Health education, Critical thinking | N | N | N | Y | Y |
| Giles 2001 | Regional | Middle school | External body | Health education | Y | N | NR | Y | Y |
| Giles 2010 | Unclear | Middle school | External body | NR | Y | NR | NR | Y | Y |
| Gonzales 2004 | Regional | High school/ Secondary school | External body | Health education | NR | N | N | Y | Y |
| Hanewinkel 2004 | International | High school/ Secondary school | External body | NR | Y | N | NR | Y | Y |
| Haruna 2018 | Local | High school/ Secondary school | External body | Health education | NR | NR | N | Y | Y |
| Hassan 2014 | Local | Elementary/ Primary school | Unclear | Humanities/ social sciences | Y | N | N | Y | Y |
| Hecht 2006 | Regional | Middle school | External body | Science and Health education | Y | N | N | Y | N |
| Heo 2021 | National | High school/ Secondary school | External body | NR | N | N | N | Y | N |
| Jacque 2016 | Regional | High school/ Secondary school | Internal body | Biology | N | N | N | Y | Y |
| Johnson 1985 | Regional | High school/ Secondary school | Unclear | Health education | N | N | N | Y | Y |
| Jones 2022 | Local | High school/ Secondary school | Internal body | NR | N | Y | N | N | Y |
| Kafewo 2008 | Local | High school/ Secondary school | External body | NR | N | NR | N | N | Y |
| Kapp 1980 | Pilot | Middle school | External body | NR | Y | N | N | Y | N |
| Kärkkäinen 2018 | Local | Elementary/ Primary school | Other | NA | N | N | N | Y | Y |

(*Continued*)

**Table 2.** (Continued)

| Study ID | Reach of intervention | Type of school | Who initiated intervention | Subject | Teacher training | Peer training | Additional materials for pupils | Assessment of intervention effect | Details of teaching method |
|---|---|---|---|---|---|---|---|---|---|
| Kärkkäinen 2019 | Local | High school/ Secondary school | External body | Health education | N | N | N | Y | Y |
| Keselman 2007 | Pilot | Middle school | Already existed | Biology | N | N | N | Y | Y |
| King 2008 | Local | High school/ Secondary school | External body | Unclear | N | N | N | Y | Y |
| Klim-Confort 2023 | Local | Middle school | Unclear | Language arts | Y | N | Y | Y | N |
| Kocken 2015 | National | High school/ Secondary school | External body | NR | Y | N | Y | Y | Y |
| Kostanjevec 2017 | Pilot | Elementary/ Primary school | External body | Home economics | N | N | N | Y | Y |
| König 2022 | National | High school/ Secondary school | External body | NA | N | N | Y | Y | Y |
| Kupersmidt 2010 | Regional | Elementary/ Primary school | External body | NR | Y | N | Y | Y | Y |
| Lakin 2008 | Local | Elementary/ Primary school | Internal body | Citizenship curriculum, Science, History, Geography, English | N | N | N | Y | Y |
| Layzer 2017 | Regional | High school/ Secondary school | External body | NR | N | Y | N | Y | Y |
| Lin 2021 | Regional | High school/ Secondary school | External body | Health education | Y | N | Y | Y | Y |
| Manesis 2022 | Local | Elementary/ Primary school | External body | NR | N | N | N | Y | Y |
| Mason–Jones 2011 | National | High school/ Secondary school | External body | Sexual education, Health education, Life orientation | N | N | Y | Y | Y |
| Maticka-Tyndale 2010 | Regional | Elementary/ Primary school | External body | Health education, Sexual education, Math, English, Critical thinking | N | N | N | Y | Y |
| Midford 2013, 2014,2016 | Regional | High school/ Secondary school | External body | Health education | Y | N | N | Y | Y |
| Marqes 2013 | Regional | High school/ Secondary school | External body | Health education | Y | Y | N | Y | Y |
| Marshman 2021 | National | High school/ Secondary school | External body | Personal health, Social education/ Health and Wellbeing | N | N | Y | N | Y |
| Mesman 2021 | Regional | High school/ Secondary school | External body | NR | Y | N | N | Y | Y |
| Modell 2023 | Regional | Middle school | External body | NR | Y | N | Y | Y | N |
| Moreno 2018 | Regional | Middle school | External body | Health education, Health Literacy, Biology, Populations statistics, Epidemiology, Social studies | N | N | N | Y | Y |
| Moreira 2010 | Local | Elementary/ Primary school | External body | Civic education, Portuguese language, Environment studies, Math | Y | N | N | Y | N |
| Neumann 1999 | Local | High school/ Secondary school | External body | Environmental health education, Math | Y | N | Y | Y | Y |

(*Continued*)

**Table 2.** (*Continued*)

| Study ID | Reach of intervention | Type of school | Who initiated intervention | Subject | Teacher training | Peer training | Additional materials for pupils | Assessment of intervention effect | Details of teaching method |
|---|---|---|---|---|---|---|---|---|---|
| **Nielsen 2023** | Regional | Unclear | External body | NR | Y | N | Y | N | N |
| **Nygard 2021** | Local | High school/ Secondary school | External body | Handicraft, Health education | Y | N | Y | Y | Y |
| **Nsangi 2017** | National | Elementary/ Primary school | External body | NR | Y | N | Y | Y | Y |
| **O'Hara 1996** | Local | High school/ Secondary school | External body | Language arts Classes | N | Y | Y | Y | Y |
| **Orsini 2019** | Regional | High school/ Secondary school | External body | NR | Y | N | N | Y | N |
| **Pacheco 1991** | Local | High school/ Secondary school | External body | English, communication skills, Health education | N | N | Y | Y | Y |
| **Palmer 2018** | Regional | Middle school | External body | Physical education | NR | N | N | Y | Y |
| **Paul 2019** | Regional | High school/ Secondary school | External body | Biology, Critical thinking | Y | N | N | Y | Y |
| **Petrie 2017** | Regional | High school/ Secondary school | External body | Health education | N | N | Y | Y | Y |
| **Perry 1989, Kelder 1995** | Regional | Middle school | External body | NR | N | Y | N | Y | Y |
| **Pieczka 2019** | Regional | High school/ Secondary school | External body | Health education, Alcohol education | N | Y | N | Y | Y |
| **Ponsford 2021** | Regional | High school/ Secondary school | External body | Sexual education | Y | N | N | Y | Y |
| **Porcu 2022** | Regional | Elementary/ primary school | External body | NR | Y | N | Y | Y | N |
| **Rajan 2017** | Regional | Middle school | External body | Health education | Y | N | N | Y | Y |
| **Reubsaet 2005** | National | High school/ Secondary school | External body | Health education | N | N | N | Y | Y |
| **Resnicow 1993** | Regional | Elementary/ Primary school | External body | Classroom generalist, Health education | Y | N | N | Y | Y |
| **Riggs 2007** | Pilot | Elementary/ Primary school | External body | NR | N | N | N | Y | Y |
| **Ridge 2002** | Regional | High school/ Secondary school and Elementary/ primary school | External body | Health education | Y | N | N | Y | N |
| **Rogow 2013** | International | High school/ Secondary school | External body | Science and humanities, Health education | Y | N | Y | Y | N |
| **Ruge 2016** | Pilot | High school/ Secondary school | Other | Health education, Nutritional education | N | N | N | Y | N |
| **Santos-Beneit 2019** | Regional | Elementary/ Primary school | External body | NR | Y | N | Y | Y | Y |
| **Seal 2006** | Local | High school/ Secondary school | External body | Health education | N | N | Y | Y | Y |
| **Schonfeld 2001** | Pilot | Pre-school, Elementary/ Primary school | Other | Health education | N | N | N | Y | N |
| **Scull 2022** | National | High school/ Secondary school | External body | Sexual health education | NR | NR | NR | Y | Y |
| **Shah 2011, 2017** | Regional | High school/ Secondary school | External body | Health education, Physical education | N | Y | NA | Y | Y |
| **Shensa 2016** | Local | High school/ Secondary school | External body | Health education, Media literacy | N | N | N | Y | Y |

(*Continued*)

**Table 2.** (Continued)

| Study ID | Reach of intervention | Type of school | Who initiated intervention | Subject | Teacher training | Peer training | Additional materials for pupils | Assessment of intervention effect | Details of teaching method |
|---|---|---|---|---|---|---|---|---|---|
| **Shinde 2017, 2020** | Pilot | High school/Secondary school | External body | NR | Y | N | NR | N | Y |
| **Simoes 2021** | National | Elementary/Primary school | External body | NR | Y | N | N | Y | Y |
| **Simon 2022** | Local | High school/Secondary school | External body | NA | Y | N | N | Y | N |
| **Timol 2016** | Regional | High school/Secondary school | External body | NR | N | Y | N | Y | N |
| **Tiwari 2020** | Local | NR | External body | NR | NR | NR | N | Y | N |
| **Türkyılmaz 2022** | Local | Elementary/Primary school | Unclear | Science | NR | N | N | Y | y |
| **Velasco 2017** | Regional | Middle school | External body | NR | Y | N | Y | Y | N |
| **Venditti 2009** | Pilot | Middle school | External body | NR | Y | N | Y | Y | Y |
| **Vieira R 2016** | Local | Elementary/Primary school | Internal body | Science | N | N | N | Y | Y |
| **Wang 2022** | Local | Unclear | External body | NR | N | N | Y | Y | Y |
| **Werle 2004** | Local | Middle school | Internal body | Health education | N | N | N | Y | N |
| **Wiist 1991** | Local | Elementary/primary school | External body | Health education | Y | Y | N | Y | Y |
| **Williams 2023** | National | Middle school | External body | NR | Y | N | Y | Y | Y |
| **Wolfe 2009, 2011** | Regional | High school/Secondary school | External body | Health education, Physical education, Sexual education | Y | N | N | Y | Y |
| **Yoon 2021** | National | High school/Secondary school | External body | Health education | Y | N | N | Y | N |
| **Zion 2021** | Unclear | Elementary/Primary school | Unclear | NR | NR | N | N | Y | Y |

*NR–not reported; NA–not applicable; Y–yes; N–no.

article), home economics (1 article), and physical education (1 article). Almost all of the 115 interventions were described as having "positive results". However, in all those cases, the evaluation concerned the entire intervention rather than single teaching methods.

## Dimensions of teaching methods used in health education

We noted a vast diversity of approaches to teaching critical thinking in health education that were tested in the included studies. To comprehensively describe this variety, we identified six dimensions that differentiated the methods based on their important characteristics listed in Fig 2.

**Central teaching component.** When we looked at the teaching methods from the perspective of the central component that organized the teaching process, we distinguished four components: practice, problem solving, exposition to stimuli, and factual content. The application of the didactical approaches in health education over five decades is presented in Table 3. While hands-on and expositional approaches prevailed in the 1980s, 1990s, and the first two decades of the 21st century, the importance of problem-solving methods has become more visible since 2011.

The teaching methods with practice as the central component provided pupils with instructions on where to gain knowledge, how to practice new skills, and how to develop new habits

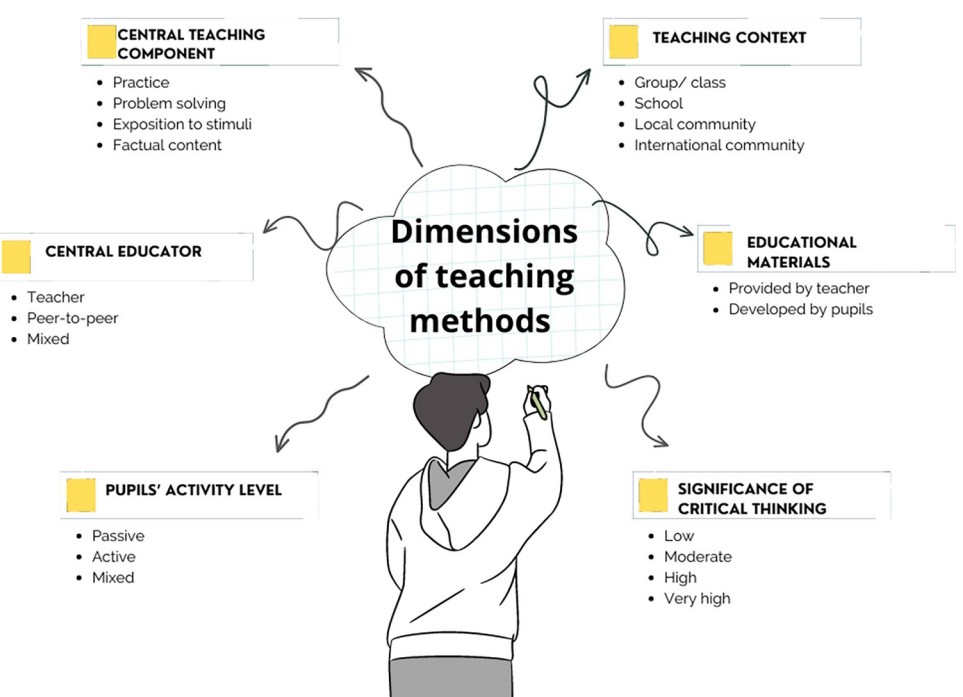

**Fig 2. Dimensions of teaching methods tested in the included studies.**

through experience. Pupils participated in or conducted practical activities that reflected the discussed issues. Typically, the practice-oriented methods were dedicated to developing either cognitive skills and emotional regulation or manual abilities and physical fitness. The former was used when fostering the skills of goal setting [77, 85, 87, 100, 137], decision-making [12, 25, 27, 29, 46, 61, 70–72, 74, 76, 77, 80, 84, 85, 89, 97, 102, 111, 120, 123, 126, 134–136, 138], stress management [85, 99], peer pressure resistance [21, 61, 80, 85, 95], emotions regulation [85, 89], peaceful conflict resolution techniques [29, 111, 139], differentiating healthy from unhealthy practice [11, 92, 123, 134, 135], assertiveness [87, 111], as well as values clarification and/or self-monitoring [46, 77, 84, 89, 120]. On the other hand, the subcategory of manual abilities and physical fitness included first aid [72], creative tasks [21, 73, 121], sports [9, 27, 60, 63, 80, 87, 106, 109], testing samples [140], daily menu composition and/or food preparation [24, 46, 96, 103, 123, 126, 134, 137, 142], project work [16, 57, 69], or making a video [14, 31, 67].

**Table 3. The central teaching component in health education interventions over five decades.**

| Decade of publication | The central teaching component | | | |
|---|---|---|---|---|
| | practice | exposition | problem solving | factual content |
| up to 1990 | 3 | 3 | 3 | 2 |
| 1991–2000 | 5 | 6 | 2 | 4 |
| 2001–2010 | 13 | 12 | 14 | 10 |
| 2011–2020 | 26 | 20 | 26 | 11 |
| from 2021 | 19 | 14 | 17 | 11 |

The number of publications calculated in rows. The colors indicate a relative number of publications calculated in the rows, with red indicating the highest and blue the lowest number.

When problem-solving is the central component of a teaching approach, pupils typically detect new knowledge and apply it in a particular situation. Pupils use "triggers" from a case study or scenario to define their own learning objectives. These methods include case study analysis [11, 13, 66, 69–72, 88, 115, 116, 130, 133], problem-based learning [89, 110, 122, 123, 125], collaborative scenario-based discussions [11, 123], storytelling [84, 110], debate [52, 91, 136], Socratic questions [52, 95], brainstorming [7, 13, 14, 64, 84, 133], and educational games [16, 17, 52, 74, 84, 85, 91, 95, 116, 118, 126, 134, 137].

Teaching methods centered on exposition offer external or internal stimuli to intensify the learning process. These methods provide pupils with an opportunity to observe particular environments and collect impressions from the external stimuli to foster the understanding of a given issue (e.g., a field trip to a sexually transmitted disease clinic [86] to university hospital to talk with medical professionals and patients [21, 140]). Alternatively, they presented posters [27, 102], video games [103, 111], videos dedicated to the health topic [98, 107, 108, 110] or allow pupils to recreate situations, reflect values, or express themselves with drama [10], role-playing [13, 26, 54, 74, 90, 95], music, and dance composition [136].

Finally, in a traditional method focusing on factual content, knowledge is delivered to pupils by means of lectures, formal presentations, or textbook work. In this approach, the teacher is the primary source of information, and pupils are recipients of information. In our analysis, factual content methods were applied in 38 (34.7%) interventions [15, 17, 19, 21, 22, 27, 46, 54, 57, 67–69, 72, 74, 80, 88, 90, 92, 94, 96, 97, 101, 102, 107, 112, 116, 120, 123, 124, 130, 132, 135–137].

In 73 interventions (63.5%), more than one component was used to reach the educational objectives. Most frequently, the authors of the intervention used all methods simultaneously [17, 69–72, 74, 94, 120, 130, 137]. They also mixed the problem-solving and practice methods [24, 30, 73, 85, 89, 91, 122, 126, 141], less often problem solving, practice methods and exposition [32, 84, 113, 140] or problem-solving and exposition [98, 106, 110, 111] and the exposition and practice methods [12, 27, 102, 103, 121]. The patterns of applying various central teaching components in the intervention addressing various health issues were grouped into seven thematic categories and presented in Table 4. While practice was central to organizing the teaching process for most health issues (more than 50% of interventions related to all health topics but SRH applied practical teaching methods), it was especially prevalent in interventions teaching about nutrition and physical activity. Problem-solving and exposition were frequently, or relatively frequently, used in interventions regarding substance use and SRH. More

**Table 4. Application of the central teaching components in interventions addressing various health issues in regard to popularity of the didactic approach in particular thematic areas.**

| Health issue | Central teaching component | | | |
|---|---|---|---|---|
| | practice | exposition | problem solving | factual content |
| psychoactive substance use | 18 | 12 | 15 | 8 |
| SRH | 11 | 16 | 17 | 11 |
| nutrition | 23 | 12 | 16 | 10 |
| public health | 12 | 11 | 8 | 9 |
| physical activity | 13 | 6 | 5 | 6 |
| somatic health | 8 | 6 | 6 | 3 |
| mental health | 9 | 5 | 5 | 3 |

The number of publications calculated in the rows. The colors indicate a relative number of publications calculated in the rows, with red indicating the highest and blue the lowest number.

than 60% of the interventions on somatic health, nutrition, and physical activity were built around more than one teaching component.

**The level of pupils' activity and central educator.** The tested teaching methods differed in terms of the level of pupils' activity. Most methods were based on the active participation of pupils and included a number of individual activities (e.g., reflection on values, goal setting, self-monitoring [87, 137]) or group activities (e.g., scenario writing [133], analyzing case and proposing a solution [29, 115]). On the other hand, in relatively few interventions, pupils were to remain passive (e.g., listening to a lecture, watching a video [25, 57, 74]). Some interventions were based on both of these forms of involvement [21, 28, 30, 46, 66, 69, 90, 98, 101, 103, 107, 111, 112, 120, 135, 136, 139, 142].

Peers play a crucial role in shaping the health behaviors of children and teenagers: they offer mutual support and serve as a role model and a trusted source of information [127]. This social dynamic was used in educational interventions across countries for over 40 years. A peer-to-peer approach was applied in 54 tested interventions [8, 10, 14–17, 21–23, 26, 28, 29, 31, 32, 46, 48, 55, 57, 58, 61, 62, 80, 81, 85, 86, 90, 92, 93, 95, 98, 99, 102, 106, 110, 112–114, 117, 119, 127, 128, 131, 133, 136, 137, 139, 140], either as a main or complementary teaching strategy. With peer-to-peer method as the main strategy, selected pupils typically participated in training for peer leaders and offered workshops, prepared presentations, or moderated discussions with other pupils [15–17, 31, 55, 57, 58, 62, 80, 81, 93, 95, 114, 117, 127, 136]. As a complementary strategy, the peer-to-peer approach was typically used at the end of the intervention. After going through the educational process, pupils created educational materials and presented them to their younger colleagues [10, 14, 21–23, 26, 46, 56, 61, 85, 90, 92, 119, 133, 137]. In 43 interventions, the teacher's role was central to the teaching process. Teachers structured the lessons, introduced content, proposed tasks, and distributed homework assignments, often according to detailed instructions [12, 13, 18–20, 22–24, 28, 29, 46, 49, 51–53, 60, 65, 67, 68, 77, 78, 80, 82, 85, 88, 90, 92, 97, 98, 101, 102, 104, 106, 112, 117, 120, 124, 125, 128, 130–134]. In every fourth intervention, teacher-centered and peer-to-peer methods were combined [12, 13, 19, 20, 22–24, 46, 53, 65, 67, 80, 85, 90, 92, 96, 117, 120, 125, 128, 131, 133]. Data on the central educator were missing in almost 37 articles.

**Educational materials.** To facilitate the learning process, every fourth of the interventions provided educational materials [7–9, 21, 22, 25, 27, 30, 32, 46, 49, 62, 69, 74, 76, 77, 81, 83, 84, 91, 93, 97, 104, 105, 107, 111, 115, 122, 132, 135, 140, 141], such as student activity books, brochures, fact sheets, activity sheets, handouts. In a number of interventions, audiovisual materials created specifically to support the teaching objectives were provided [20, 74, 90, 115].

In 30% of the interventions, the learning process resulted in pupils creating some artefacts. Some of those creative works served as a souvenir and were supposed to remind pupils of the health issue they were taught about [125, 137]. Other works had additional educational purposes, such as a poster exhibition [23, 28, 29, 31, 32, 47, 73, 77, 83, 86, 107], creating a cartoon about the rational use of medicines [12], shooting a video about the process of making reusable sanitary cloth pads [14], developing an educational website on cancer prevention for children that was posted on the website of the Yale Cancer Center [131]. In some interventions, children prepared and consumed foods with certain nutritional values (e.g., low-fat, high-fiber products [77, 87]) or foods from different cultural contexts [8].

In one in three interventions, computer, internet, or other technological tools were used to support the educational process. The application of teaching methods was typically supported by internet search [11, 13, 22, 50, 60, 66, 73, 75, 97, 100, 102, 103, 105, 108, 115, 122, 131, 132], creating presentations [20, 22, 29, 30, 46, 47, 54, 74, 111, 140], communicating or analyzing social media [7, 8, 11, 13, 17, 20, 30, 31, 108, 122], using applications, both those generally

available, i.e. interactive web-based quiz and those developed for the intervention [12, 88, 91, 99, 102, 106, 110, 115, 118], or computer games [84, 103, 107, 111, 118, 134].

**Teaching context.** Within the model of health promoting schools, introduced by the World Health Organization after the release of the Ottawa Charter during the first International Conference on Health Promotion in Ottawa, Canada, in 1986, the socio-ecological perspective on health education was applied in schools [23]. As a result, a number of educational interventions on health involved activities engaging the whole school community [10, 15, 16, 21, 24, 26, 27, 32, 46, 57, 78, 81, 86, 87, 96, 99, 100, 103, 109, 113, 119, 124, 137, 141, 142] or even a broader local community [45, 63, 66, 69, 89, 93, 102, 103, 112, 116, 131, 133, 141, 143, 145], and not just standard classroom teaching. In some studies, not only was the pupil-teacher relationship explored, but also contacts with other social actors were arranged. Twenty-three interventions engaged pupils' parents and caregivers [23, 27, 28, 30, 32, 46, 48, 52, 59, 72, 77, 78, 87, 93, 94, 97, 102, 111, 113, 115, 119, 126, 128]; 12, external experts and scientists [12, 16, 17, 19, 73, 77, 86, 90, 103, 115, 118, 124, 132, 135]; and 8, other social actors [8, 21, 23, 31, 99, 117, 120, 138] such as school administrators, local leaders, or school nurses. The involvement of parents in some interventions ranged from providing information materials [78] to providing technical support (e.g., parents who were farmers provided soil for planters [126]). Parents were also involved through shared activities [97], or they were offered to participate in classes on communicating personal and family's values about sexuality to teenagers [52, 94, 115, 128], or they received newsletters or magazines with health information, heart-healthy recipes, and hands-on activities to do at home [46, 87, 93].

**Significance of critical thinking.** The stage of eligibility criteria assessment showed that critical thinking was included only in a small proportion of health education interventions for children and adolescents. However, the interventions described in the included publications varied with regards to: 1) the methods applied to develop critical thinking skills; and 2) the extent to which they provided details on the teaching process. Based on the information and additional materials provided in the articles, we used those two parameters to evaluate the significance of critical thinking in the tested interventions on a four-point scale (low, moderate, high, and very high significance) (Table 5).

Most interventions (42 articles, 36%) described only one method addressing critical thinking and failed to provide details of the activities. In these interventions, critical thinking was classified as having a low level of significance. The most common approaches reported by the authors were group discussions or debates [7, 9, 15, 20, 26, 47, 53, 59, 62, 63, 69, 76, 93, 109, 117], Socratic discussions [52, 95], question boxes [94, 124], unspecified decision-making exercises [23–25, 49, 55, 65, 68, 74, 77, 85, 97, 112, 127, 134], or reflection activities [118]. The low significance of critical thinking teaching methods was noted in interventions from all decades.

**Table 5. Significance of critical thinking in educational interventions addressing different health issues.**

| Decade of publications | Level of significance of critical thinking | | | |
|:---:|:---:|:---:|:---:|:---:|
| | **low** | **moderate** | **high** | **very high** |
| up to 1990 | 1 | 0 | 2 | 1 |
| 1991–2000 | 4 | 3 | 1 | 0 |
| 2001–2010 | 16 | 7 | 3 | 5 |
| 2011–2020 | 17 | 11 | 9 | 6 |
| from 2021 | 4 | 9 | 12 | 3 |

The number of publications calculated in the rows. The colors indicate a relative number of publications calculated in the rows, with red indicating the highest and blue the lowest number.

**Table 6. Significance of critical thinking in educational interventions addressing different health issues.**

| Health issue | Level of significance of critical thinking | | | |
|---|---|---|---|---|
| | low | moderate | high | very high |
| psychoactive substance use | 13 | 10 | 7 | 1 |
| SRH | 13 | 8 | 7 | 3 |
| nutrition | 8 | 5 | 9 | 5 |
| public health | 6 | 6 | 3 | 6 |
| physical activity | 7 | 1 | 3 | 2 |
| somatic health | 4 | 1 | 5 | 1 |
| mental health | 4 | 3 | 5 | 1 |

The number of publications calculated in the columns. The colors indicate a relative number of publications calculated in the columns, with red indicating the highest and blue the lowest number.

Critical thinking educational methods were most commonly applied in interventions regarding substance use and SRH (Table 6). Half of the intervention addressing physical activity and more than 40% addressing psychoactive substance use and SRH demonstrated a low significance of critical thinking.

In 21 interventions, more than one method stimulating critical thinking was listed. Critical thinking in these interventions was classified as having moderate significance. However, activities for developing critical thinking skills constituted a small part of a broader educational program or the articles did not provide details suggesting otherwise [12, 17, 21, 29, 48, 50, 54, 57, 60, 75, 82–84, 86, 92, 99, 100, 108, 110, 114, 119, 121, 128, 132, 139, 143, 144]. Apart from discussion or decision-making exercises, these interventions typically involved other methods facilitating critical thinking, such as situational role playing, problem-solving, participation in developing educational activities on health, designing wall magazines, assessing individual or community health resources, analyzing media information, and solving case studies For about 30% of the interventions addressing psychoactive substance use and SRH teaching critical thinking was of a moderate importance.

The interventions classified as showing a high or very high significance of critical thinking included multiple teaching methods stimulating critical thinking skills and provided a detailed description of the whole educational process, a relationship between the teaching objectives and applied teaching methods, and how they were translated into specific learning activities, materials, and outcomes.

Twenty-seven interventions characterized by high significance of critical thinking [11, 13, 14, 22, 28, 30, 32, 64, 67, 70–73, 80, 89, 98, 102, 105, 106, 111, 113, 120, 122, 123, 126, 131, 140–142] discussed a broader scope of health literacy skills, with critical thinking being only one of those skills. On the other hand, interventions with a very high level of significance [8, 10, 18, 31, 61, 88, 96, 115, 125, 130, 133, 135, 137, 138] were dedicated to critical thinking and comprehensively addressed a set of skills involved. Reporting on educational interventions that approached critical thinking in a more complex manner became more common after 2000. Critical thinking gained more significant coverage in more than half of the interventions focused on nutrition (52%). We observed high or very high significance of critical thinking in interventions teaching about somatic health (46%), physical activity (46%) and public health (45%) (Table 6).

High and very high significance was demonstrated especially for interventions that incorporated problem-solving as opposed to those with practice as the central component. The

**Table 7. Significance of critical thinking in educational interventions intersected with categorization regarding of central teaching component.**

| Central teaching component | Level of significance of critical thinking | | | |
|---|---|---|---|---|
| | **low** | **moderate** | **high** | **very high** |
| practice | 23 | 16 | 18 | 8 |
| exposition | 16 | 15 | 13 | 10 |
| problem solving | 16 | 12 | 22 | 12 |
| factual content | 12 | 11 | 9 | 5 |
| mixed | 19 | 16 | 14 | 10 |

The number of publications calculated in the rows. The colors indicate a relative number of publications calculated in the rows, with red indicating the highest and blue the lowest number

latter interventions were characterized mainly by low significance of methods addressing critical thinking (Table 7).

Examples of the most interesting interventions in which critical thinking had high or very high significance are described in Table 8.

## Discussion

### Summary of the main results

Our scoping review demonstrated a large variety of educational interventions regarding health issues over time and across continents. The interventions reported in the included articles focused mainly on lifestyle-related health issues, which reflect the dynamic changes in the discourse on the health of children and adolescents as well as in the priorities of health prevention programs [145–147]. Healthy lifestyle interventions implemented before 2011 typically aimed at developing knowledge, skills, and/or attitudes related to substance use, SRH, and broader problems of public health. Subsequent interventions seem to reflect the more recent conceptualization of healthy lifestyle in relation to an increase in obesity in children [148], as they additionally cover habits linked to nutrition and physical activity. More specific aspects of individual health, such as particular somatic or mental health disorders, seem to be receiving more attention in health education interventions in 21st century. The regional dynamics of the coverage of health topics, as observed in our review, can be explained by various regional health challenges and local socio-cultural determinants of health.

A similar diversity was noted in the teaching methods applied in the interventions studied over the period of 40 years. While older interventions (before 2001) primarily focused on exposing students to external or internal stimuli, delivering factual content or practical activities to promote health behaviors, the more recent interventions design the educational process around problem-solving tasks. The teaching methods used in the interventions addressing nutrition and physical activity were mostly oriented towards developing practical skills, while those applied in the interventions addressing sexual health or substance use emphasized problem-solving skills. Mixing those various components was a strategy applied in interventions addressing all thematic areas.

In some interventions, the teaching process was accompanied by various types of educational materials, and sometimes pupils created educational artefacts themselves. Most teaching methods used in the studied interventions encouraged pupils to actively participate in the learning process, express their opinions in writing, or develop various types of educational materials. Such approaches facilitate the integration of knowledge, skills, and essential components of attitudes. Some articles tested interventions that engaged peer educators in promoting

**Table 8. Interventions with high and very high level of significance of critical thinking in teaching methods addressing a given health issue.**

| Health issue | Teaching methods | Description of the intervention |
|---|---|---|
| SRH: HIV/AIDS prevention | Problem solving | Pupils were asked to write a response to a teenager's question about her risk of contracting a sexually transmitted disease from her boyfriend. Small groups of pupils assumed the role of an HIV clinic counsellor. After an in-depth analysis of her situation and identification of her misconceptions about HIV, pupils were supposed to write down information to improve her understanding [133]. |
| SRH | Exposition and problem-solving | *If I Were Jack* was a relationships and sexuality education program resource that focused on young men and unintended pregnancy. It was based on an interactive video drama that told the story of Jack, a teenager who had just found out that his girlfriend is unexpectedly pregnant. Pupils were encouraged to discuss Jack's situation as well his and his girlfriend's options and decisions. The education program was designed to promote critical thinking about social pressures that normally situated teenage pregnancy and to go beyond the gender stereotypes surrounding teenage pregnancy [115]. |
| Substance-use prevention | Problem-solving | The intervention consisted of three components. In the first component, pupils shared their own beliefs about cigarette smoking and confronted them with the knowledge of their peers as well as expert knowledge. Then, through role-playing, pupils learned to resist pressure (from peers, the media). The third component was about decision-making and commitment, where pupils integrated all of the information and were asked to consider the social consequences of smoking in their own social environment. Each pupil then made a decision of whether to smoke or not, along with providing the main reason. The decision, along with the reasons, was announced in front of classmates [61]. |
| Nutrition | Practical and problem-solving | Pupils debated the fictive cases brought up in the blogs provided by the teacher and applied their evidence-based knowledge to solve the nutritional dilemma presented in the blogs. They explained and argued the kind of guidance they had given to their cases, and then, the whole class discussed the cases and the adequacy of prescribed instructions [11]. |
| | Problem-solving | *The shopping bag game* involved selecting different foods and justifying the choices made. The children were presented with a selection of different food items, e.g.: vegetables, yoghurt, cheese, and eggs. Each product contained a ticket with information such as the cost of the food, its country of origin, how far it has travelled, and whether it is organic or nonorganic. The children shopped by selecting product tickets. At the end of the game, they had to say what influenced their choice [126]. |
| Physical activity | Practical and problem-solving | The intervention that combined a number of activities, including those directed at assessing one's physical activity and diet and proposing solutions for oneself, others, and the local environment. Among other things, the students used the knowledge they gained in finding solutions and advising their peer, Calvin, from the case study, who would like to return to playing basketball after years of unhealthy lifestyle. They used pedometers to check their activity throughout the day, and analyzed facilities that encourage a sedentary lifestyle. The culmination of the intervention was the development of an artifact that would help their peers, parents, school community or the community at large change their current environment or navigate it to make healthy food and activity choices [96]. |

(*Continued*)

**Table 8.**  (Continued)

| Health issue | Teaching methods | Description of the intervention |
|---|---|---|
| Mental health | Exposition and problem-solving | The intervention included three phases: readiness, instructional, and application. In the readiness skills phase, pupils were trained through role-playing activities to actively listen to others and to self-control. Pupils received positive and/or corrective feedback and were guided to recognize needs and feelings in themselves and others, and to develop a sense of responsibility as a group member. During the instructional phase, pupils developed the steps required for social problem-solving and decision-making and finally trying out the solutions in a safe environment [64]. |
| Somatic health | Problem-solving | During class on respiratory system and health, pupils used their knowledge on research in science and practiced communication skills in expressing agreement or disagreement and considering reasons in favor of the opposite point of view and refute them to wrote an argumentative essay entitled "Do you agree or disagree with the use of images of people smoking on television?" [125]. |
| Public health | Problem-solving, practical, exposition | Learning about Danish and Kenyan food culture in the context of health inequalities, pupils from two countries used letters and online communicators to get to know each other and. They shared their daily experiences and typical food products to understand interdependence between people and nations as well as differences in lifestyle and health behaviours [8]. |
|  | Exposition | Individuals with lived experience of violence from the Veterans Education Project shared their stories with pupils, who were then instructed to write a response to open-ended questions for two minutes. The questions were designed to be neutral and to assist students in organizing their thoughts: *What was your response to the story? What was the main message of the story? What were the storyteller's attitudes about violence? How did these attitudes change as a result of the storyteller's experiences? What did you like about the story? What did you dislike?* Pupils responded in a free writing format [18]. |

healthy choices, presenting useful skills, and explaining health information. While most of the available evidence suggests the effectiveness of peer-to-peer teaching in higher education [149, 150], a recent scoping review of studies on peer education in health interventions for adolescents revealed that involving peer-to-peer education may be a promising strategy for health improvement also on lower educational levels [151]. The way of shaping health behaviors in the included interventions focused not only on expanding the knowledge of individual pupils as well as training their health-related skills, but also encompassed the broader social context of pupils: their families, local communities, or intercultural contacts. Moreover, in some interventions, pupils met medical professionals, patients and their caregivers, or external experts and scientists, sometimes in their work setting.

In summary, there is evidence to suggest that peer-to peer interaction is one of the teaching strategies related to student gains in critical thinking. Therefore, leaving the role of the central educator to pupils and designing interventions that engage pupils in individual and group activities (such as problem solving, developing educational materials or artefacts) are possibly those dimensions of the teaching methods that offer greatest benefits in terms of learning critical thinking skills.

## Importance of critical thinking in health education of children up to high school

The extent to which the included interventions covered critical thinking skills varied widely. This heterogeneity is associated with the year of the publication and the dynamics of

pedagogical discourse. The growing demands of the contemporary information society [22] and changing public health challenges in the past four decades has resulted in a growing appreciation of teaching critical thinking. The increase in the complexity of integrating critical thinking into educational interventions is particularly evident in the publications released from 2021.

## Strengths and limitations

To our best knowledge, this is the first study to comprehensively review the existing literature on the teaching methods for critical thinking in the health education of children up to high school. The review was conducted by an interdisciplinary team and was based on an extensive literature search including all types of research from all continents.

Our review also has some limitations. As our search was performed in 20 September 2023, there is a considerable disproportion in the number of articles between decades, with fewer articles categorized as those published from 2021 as compared with the earlier decades. Moreover, the studies and interventions included in the review were highly heterogenous, and the description of some teaching methods was not satisfactory, limiting possibility to replicate them. Some of the included studies only listed the teaching methods without any additional information. Developing reporting checklist for health education interventions in school context- such as to TIDieR checklist [152] available for interventions in general or GREET [153] for evidence-based practice educational interventions, may improve future reporting and replicability of such interventions. Moreover, as we were interested in the educational programs stably functioning in the school setting and engaging school-based actors, we excluded interventions that were implemented only by external educators, external leaders, medical school students, or medical professionals. Future studies should map the methods applied in extracurricular interventions. Finally, we included only articles in English; thus, we potentially missed out on studies published in other languages.

## Conclusions

Our review showed that health education interventions in children and adolescents usually did not address the development of critical thinking skills in a comprehensive manner. Interventions in which critical thinking had high and very high significance applied mainly problem-solving methods and involved pupils' activity. The evidence on the effectiveness of the teaching methods that develop critical thinking skills is limited because most articles failed to provide detailed information on the teaching methods or did not examine their effects. Therefore, to facilitate further research in this field, we recommend that the teaching strategies used in the interventions are described in greater detail and that the effectiveness of individual teaching methods is assessed and reported. The development of a reporting checklist to describe health education interventions is warranted.

## Supporting information

**S1 Table. Preferred Reporting Items for Systematic reviews and Meta-Analyses extension for Scoping Reviews (PRISMA-ScR) checklist.**
(DOCX)

**S2 Table. Search strategies.**
(DOCX)

**S3 Table. Characteristics of the included studies.**
(DOCX)

## Acknowledgments

We thank dr Magdalena Koperny for creating the search strategy.

## Author Contributions

**Conceptualization:** Anna Prokop-Dorner, Aleksandra Piłat-Kobla, Magdalena Ślusarczyk, Maria Świątkiewicz-Mośny, Natalia Ożegalska-Łukasik, Małgorzata M. Bała.

**Data curation:** Anna Prokop-Dorner, Aleksandra Piłat-Kobla, Magdalena Ślusarczyk, Maria Świątkiewicz-Mośny, Natalia Ożegalska-Łukasik, Aleksandra Potysz-Rzyman, Marianna Zarychta, Albert Juszczyk, Dominika Kondyjowska, Agnieszka Magiera, Małgorzata Maraj, Dawid Storman, Sylwia Warzecha, Paulina Węglarz, Magdalena Wojtaszek-Główka, Wioletta Żabicka, Małgorzata M. Bała.

**Formal analysis:** Anna Prokop-Dorner, Aleksandra Piłat-Kobla, Małgorzata M. Bała.

**Funding acquisition:** Anna Prokop-Dorner, Aleksandra Piłat-Kobla, Magdalena Ślusarczyk, Maria Świątkiewicz-Mośny, Natalia Ożegalska-Łukasik, Małgorzata M. Bała.

**Investigation:** Anna Prokop-Dorner, Aleksandra Piłat-Kobla, Małgorzata M. Bała.

**Methodology:** Anna Prokop-Dorner, Aleksandra Piłat-Kobla, Małgorzata M. Bała.

**Project administration:** Maria Świątkiewicz-Mośny, Małgorzata M. Bała.

**Resources:** Maria Świątkiewicz-Mośny, Małgorzata M. Bała.

**Software:** Anna Prokop-Dorner, Aleksandra Piłat-Kobla, Małgorzata M. Bała.

**Supervision:** Małgorzata M. Bała.

**Visualization:** Anna Prokop-Dorner, Aleksandra Piłat-Kobla.

**Writing – original draft:** Anna Prokop-Dorner, Aleksandra Piłat-Kobla.

**Writing – review & editing:** Anna Prokop-Dorner, Aleksandra Piłat-Kobla, Magdalena Ślusarczyk, Maria Świątkiewicz-Mośny, Natalia Ożegalska-Łukasik, Aleksandra Potysz-Rzyman, Marianna Zarychta, Albert Juszczyk, Dominika Kondyjowska, Agnieszka Magiera, Małgorzata Maraj, Dawid Storman, Sylwia Warzecha, Paulina Węglarz, Magdalena Wojtaszek-Główka, Wioletta Żabicka, Małgorzata M. Bała.

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
