## [Decision Letter · Decision Letter 0]

21 May 2024

PONE-D-24-05236Teaching methods for critical thinking in health education of children up to high school: a scoping reviewPLOS ONE

Dear Dr. Prokop-Dorner,

Thank you for submitting your manuscript to PLOS ONE. After careful consideration, we feel that it has merit but does not fully meet PLOS ONE’s publication criteria as it currently stands. Therefore, we invite you to submit a revised version of the manuscript that addresses the points raised during the review process. Congratulations for your work! There are few necessary adjustments, please find below the details from our reviewers. We are looking forward for your upgraded version. Please submit your revised manuscript by Jul 05 2024 11:59PM. If you will need more time than this to complete your revisions, please reply to this message or contact the journal office at plosone@plos.org. Please include the following items when submitting your revised manuscript:A rebuttal letter that responds to each point raised by the academic editor and reviewer(s). You should upload this letter as a separate file labeled 'Response to Reviewers'.A marked-up copy of your manuscript that highlights changes made to the original version. You should upload this as a separate file labeled 'Revised Manuscript with Track Changes'.An unmarked version of your revised paper without tracked changes. You should upload this as a separate file labeled 'Manuscript'.If applicable, we recommend that you deposit your laboratory protocols in protocols.io to enhance the reproducibility of your results. Protocols.io assigns your protocol its own identifier (DOI) so that it can be cited independently in the future. For instructions see: https://journals.plos.org/plosone/s/submission-guidelines#loc-laboratory-protocols. Additionally, PLOS ONE offers an option for publishing peer-reviewed Lab Protocol articles, which describe protocols hosted on protocols.io. Read more information on sharing protocols at https://plos.org/protocols?utm_medium=editorial-email&utm_source=authorletters&utm_campaign=protocols.

We look forward to receiving your revised manuscript.

Kind regards,

Bogdan Nadolu, Ph.D.

Academic Editor

PLOS ONE

“This work is the result of research project Diagnosis and developing health capital - Health Literacy of primary school students (Project no. UMO-2020/39/B/HS6/00977) funded by the National Science Centre.”

Reviewers' comments:

Reviewer's Responses to Questions

**Comments to the Author**

1. Is the manuscript technically sound, and do the data support the conclusions?

Reviewer #1: Yes

Reviewer #2: Yes

2. Has the statistical analysis been performed appropriately and rigorously? 

Reviewer #1: Yes

Reviewer #2: I Don't Know

3. Have the authors made all data underlying the findings in their manuscript fully available?

Reviewer #1: Yes

Reviewer #2: Yes

4. Is the manuscript presented in an intelligible fashion and written in standard English?

Reviewer #1: Yes

Reviewer #2: Yes

5. Review Comments to the Author

Reviewer #1: The scoping review followed JBI and PRISMA, the gold standards, for this review paper. It is beautifully written with no grammatical or editorial mistakes. It is interesting to see that the authors used MASQDA to conduct the qualitative synthesis and reported the results based on the six dimensions. The methodology is sound. The results are clear and well-organized. The discussion is not as strong as Methods and Results but acceptable. I do have a question regarding 1056 studies. When I calculated the numbers listed in the first paragraph of Results, the total was 1053.

Reviewer #2: Thank you for the opportunity to review this scoping review. I am screening from the lens of an information specialist trained on comprehensive and transparent literature search methods.

I want to applaud the authors for their database selection, reporting the dates when searches were last executed, and sharing their entire search strategies in the supplementary materials. For the most part, they are reproducible. To ensure reproducible search methods, the authors will want to indicate which platforms were used for all databases. This was done with some (e.g., Ovid Medline) but not all (e.g., EMBASE via ?).

I have a concerns about the comprehensiveness of the literature search. For some of the databases, there were only subject heading searches performed without targeting other metadata fields like titles, abstracts, and author keywords. I worry that some studies could have been missed in this approach. Could you justify this choice?

6. PLOS authors have the option to publish the peer review history of their article (what does this mean?). If published, this will include your full peer review and any attached files.

Reviewer #1: No

Reviewer #2: No

---

## [Author Response · Author response to Decision Letter 0]

13 Jun 2024

Thank you, Reviewer 1, for taking the time to review our work. We appreciate your kind comments and positive feedback.

Thank you for paying our attention to our calculation of the studies sought for retrieval. We have verified it and cleared it out in the text. 

Our calculation looks as follows: the total number of the studies sought for retrieval is 1056, of which:

- 938 were excluded due to 10 different reasons 

- 3 potentially eligible studies were identified as ongoing 

- 115 eligible studies were included. 

We revisited the explanation of this calculations that we provided initially in the manuscript and cleared it out. Now it states:

We identified 118 eligible studies, of which 3 were still ongoing [43-45]. Finally, we included 115 completed studies.

Thank you, Reviewer 2, for your positive feedback on our work and for your detailed comments. We appreciate the time and efforts you have put into reviewing our manuscript. Below we respond to your remarks.

1. Thank you for the comment regarding the platforms.

We used the following platforms to search the databases:

- Embase from Elsevier

- WoS from Clarivate

- Medline from Ovid

- ERIC from EBSCO

- PsycArticles from EBSCO

- CINAHL from EBSCO

- Proquest from Proquest central

We added to the missing details to the Supporting information 2.

2. As far as comprehensiveness of the literature search, we have tested several strategies to choose the option that would be optimal and feasible for research question. In the main databases (Medline and Embase) we used a comprehensive approach, and we used MesH Tesaurus terms and fields. In Medline via Ovid .mp which means that database search title, abstract, original title, name of substance word, subject heading word, protocol supplementary concept word, rare disease supplementary concept word, unique identifier). In Embase we used an explode option (major topic), index term and some keywords we searched in all fields ('teaching' OR 'curriculum' OR 'education’/ 'critical thinking' OR 'thinking'). Web of Science Core Collection employs no controlled vocabulary or thesaurus, so we used abstract, title, author keywords. In CINHAL database we used Word in Subject Heading, Exact Subject Heading and for some keywords, such as "critical thinking", we used all text search. For some terms in Proquest Central, like “Health education”, we used the main subject option because when we had used a broader search without restrictions, we have got huge number of results, and when we provide initial selection, most of the records were irrelevant. Additionally, even after narrowing down the search fields in databases like ProQuest and CINHAL, a high number of results persisted and was screened. During the selection process, many records were still excluded. 

We believe that searching across multiple databases minimizes the risk of overlooking crucial publications pertinent to the research problem under analysis.

---

## [Editor Report · Decision Letter 1]

1 Jul 2024

Teaching methods for critical thinking in health education of children up to high school: a scoping review

PONE-D-24-05236R1

Dear Dr. Prokop-Dorner,

We’re pleased to inform you that your manuscript has been judged scientifically suitable for publication and will be formally accepted for publication once it meets all outstanding technical requirements.

Kind regards,

Bogdan Nadolu, Ph.D.

Academic Editor

PLOS ONE
---

## [Editor Report · Acceptance letter]

7 Jul 2024

PONE-D-24-05236R1 

PLOS ONE

Dear Dr. Prokop-Dorner, 

I'm pleased to inform you that your manuscript has been deemed suitable for publication in PLOS ONE. Congratulations! Your manuscript is now being handed over to our production team.

Kind regards, 

on behalf of

Dr. Bogdan Nadolu 

Academic Editor

PLOS ONE